# PROACTIVE POLICING AS REINFORCMENT LEARNING

**Dawson Kinsman**
Department of Mathematics and Statistics
University of Michigan-Dearborn
Dearborn, MI 48128, USA
tiananw@umich.edu

**Tian An Wong**
Department of Mathematics and Statistics
University of Michigan-Dearborn
Dearborn, MI 48128, USA
dkinsman@umich.edu

## ABSTRACT

Recent analyses of predictive policing have shown the inherent biases in such systems. We show that the models considered in fact apply to proactive policing in general, which can be also viewed as a reinforcement learning system, and thus may also lead to over-policing.

## 1 INTRODUCTION

Predictive policing algorithms, most notably PredPol, were developed about a decade ago, and have more recently been subject to scrutiny. Most notably, studies such as Lum & Isaac (2016); Ensign et al. (2017; 2018); Akpinar et al. (2021) show empirically and theoretically that negative feedback loops occur in predictive policing algorithms, leading to over-policing. Whereas these critiques have naturally been understood from the perspective of algorithmic bias, it has been argued by social scientists that policing itself is biased *without* the help of machine learning algorithms (e.g, Kahn & Martin (2016)). We argue that proactive policing in general can be viewed as a reinforcement learning system, and as such the models used in analyses of algorithmic predictive policing can be adapted to show that bias generally occurs in non-algorithmtic policing. This provides a clear framework showing the continuity between algorithmic and non-algorithmic forms of policing, and suggests ways of analyzing non-algorithmic systems *as* generalized learning algorithms. Moreover, this also has public health implications as the multiplier effect can lead to a higher risk of encountering police violence, particularly as these often occur in communities of color.

## 2 PROACTIVE POLICING AS REINFORCEMENT LEARNING

Algorithmic policing extends more rudimentary forms of policing based on crime statistics such as NYPD's CompStat program, adopted in 1995 Shapiro (2017). The simple logic, based on the broken-windows policing paradigm, is that policing resources should be invested more in areas with higher crime rates. (We leave aside the important question of *which* crimes are considered.) Importantly, Ensign et al. (2017; 2018) define *predictive policing* as:

**Definition 1 (Predictive policing)** *Given historical crime incident data for a collection of regions, decide how to allocate* patrol officers *to areas to detect crime.*

Notice there is nothing in this definition that is specifically algorithmic, which we argue is really an important point. This should be compared with the more general definition of *proactive policing*, defined in National Academies of Sciences et al. (2018) as policing strategies that have as a goal the prevention or reduction of crime and that are not reactive in terms of focusing primarily on uncovering ongoing crime or on investigating or responding to crimes once they have occurred. As such, we can expand the definition of Ensign et al. (2017; 2018) simply as follows:

**Definition 2 (Proactive policing)** *Given historical crime incident data for a collection of regions, decide how to allocate* policing resources *to areas to detect crime.*

That is, we generalise "patrol officers" to "policing resources," to account for other forms of surveillance such as gunshot detection and cameras, often equipped with license plate readers and facial

recognition systems. The key observation is that it is not only patrol officers that detect crime, but other surveillance systems also that increase the rate of crime detection. Invariably, such systems lead to increased police presence, so the final result of increased policing holds. (Indeed, it can be inferred from Akpinar et al. (2021) that increased crime detection leads to increased policing.)

Ensign et al. (2017) model predictive policing with a partial monitoring framework in reinforcement learning. A partial monitoring problem is a tuple $P = (A, Y, H, L)$ where $A = \{a_1, \ldots, a_n\}$ is a set of $n$ actions, $Y = \{y_1, \ldots, y_m\}$ a set of $m$ outcomes (adversary actions), $H : A \times Y \to \Sigma$ a feedback matrix valued in $\Sigma$ the set of information (signals) that the learner can receive, and $L : A \times Y \to \mathbb{R}$ a loss function (Cesa-Bianchi & Lugosi, 2006, Chapter 6). In partial monitoring, the loss $L$ is not known to the learner after an action is taken. Ensign et al. (2017) prove regrets bounds on the order of $O(T^{2/3})$ against worst-case adversaries in a two-neighborhood game, and furthermore prove in Ensign et al. (2018) the existence of runaway feedback loops via a Pólya urn model. These models immediately generalize from the set $A$ of patrolling decisions to the investment of proactive policing resources that claim to detect crime, and the preceding results apply (see Appendix A.1–A.3 for details). In other words, proactive policing may also lead to over-policing.

As a demonstration of this principle using synthetic data, we implement a basic model on two locations (Appendix B).

## 3  DISCUSSION

This subtle paradigm shift should not be underestimated: a quick review of the predictive policing literature concerns predictive policing *algorithms*, where in fact the general study of proactive policing, algorithmic or not, is often treated in practice as a reinforcement learning problem, and studies used to analyze predictive policing as black box systems should also be applied to proactive policing. Because predictive policing systems encode historical bias (notice that the regret bound $O(T^{2/3}$ accumulates with time), it may be useful to (carefully) retroactively apply advances in reinforcement learning bias measures to proactive policing. See Appendix A.4 for added discussion on potential debiasing methods and their limitations.

Having established that proactive policing generates runaway feedback loops in general with small and big data, we turn to the question of 'crime' itself. Qualitatively, police violence is a different kind of crime and should of course be studied differently. Nonetheless, multiple studies suggest police violence spreads across police misconduct networks as a social contagion. Notably, whereas Ouellet et al. (2019); Simpson & Kirk (2022) rely on police misconduct co-complaint networks, Quispe-Torreblanca & Stewart (2019) uses broader data involving officer peer networks and give evidence for the contagion effect of police misconduct.

To return to a reinforcement learning framework, we note that Ensign et al. (2017) describe both predictive policing and recidivism prediction as partial monitoring problems, whereas Khorshidi et al. (2020) adapt recidivism risk forecasting methods to predict police excessive use-of-force. We also note that Mohler in Khorshidi et al. (2020) used the exact same Hawkes process to model the spread of police violence across misconduct networks as he used for PredPol Lum & Isaac (2016), suggesting that violence spreads just as well across police networks as through criminal networks.

When applied to a scenario of over-policing, this results in an increased probability that police misconduct, and in particular police violence, will occur in a given neighborhood with higher likelihood. This should be understood as multiplier effect on police violence as a result of the negative feedback loops and warrants further study.

## 4  CONCLUSION

Examining the mathematical models of predictive policing, we find the same can be adapted to proactive policing, thus understood as a reinforcement learning or partial monitoring system. Together with the social contagion effects police misconduct, we conclude that proactive policing can lead to over-policing, which may in turn produce a compounding effect of police violence in over-policed neighborhoods. Precisely tuned fairness penalties may not be as effective or practical as broader violence intervention in communities at risk of violence, both civilian and police, and requires a multidisciplinary approach.

ACKNOWLEDGEMENTS

The second author was partially supported by NSF grant DMS-2212924.

URM STATEMENT

The authors acknowledge that at least one key author of this work meets the URM criteria of ICLR 2023 Tiny Papers Track.

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

## A   THE CRIME MODELS

### A.1   CRIME INCIDENCE AS A HAWKES PROCESS

In this Hawkes process model of crime, such as in Khorshidi et al. (2020), policing areas are discretized into square boxes. The probabilistic rate of events in box $n$ at time $t$ is defined to be

$$\lambda_n(t) = \mu_n + \sum_{t_n^i < t} \theta \omega e^{-\omega(t - t_n^i)},$$

where $t_n^i$ are the times of events in box $n$ in the history of the process. The background rate $\mu$ is a (nonparametric histogram) estimate of a stationary Poisson process. The expectation, or E-step, sets

$$p_n^{ij} = \frac{\theta \omega e^{-\omega(t_n^i - t_n^j)}}{\lambda_n(t_n^j)}, \qquad p_n^j = \frac{\mu_n}{\lambda_n(t_n^j)},$$

where $\theta \omega e^{-\omega t}$ is called the triggering kernel that models near-repeat or contagion effects in crime data. The maximization, or M-step, sets

$$\omega = \frac{\sum_n \sum_{i<j} p_n^{ij}}{\sum_n \sum_{i<j} p_n^{ij}(t_n^j - t_n^i)}, \quad \theta = \frac{\sum_n \sum_{i<j} p_n^{ij}}{\sum_n \sum_j 1}, \quad \mu = \frac{\sum_n \sum_j p_n^j}{T},$$

where $T$ is the length of the time window of observation. Policing resources are then allocated according to the expected rate of crime at each box. As more police are sent to each box, the contagion of police misconduct on the other hand spreads with higher probability in more highly police areas.

### A.2   THE URN MODEL OF PROACTIVE POLICING

Ensign et al. Ensign et al. (2018) model develop an urn model of predictive policing showing that runaway feedback loop occurs in predictive policing. The use of urns is a common frame-work in reinforcement learning. In the generalized Pólya urn model, an urn contains balls of two colors, say red and black. At each time step, a ball is drawn, and based on its color a number of balls are replaced. If red, we add $a$ red and $b$ black balls; and if black we add $c$ red and $d$ black balls. This is represented by the replacement matrix

$$\begin{pmatrix} a & b \\ c & d \end{pmatrix},$$

where the basic case is when $a = d = 1$ and $c = b = 0$.

As a toy model for predictive policing via allocation of patrol officers, $A$ and $B$ are two policing areas, and the goal is to distribute police officers according to the proportion of crime in each area. Let $d_A$ be the rate at which police in $A$ discover crimes, $r_A$ the rate at which crimes are reported in $A$, and $w_d, w_r$ the respective weights such that $w_d + w_r = 1$ and $w_d d_A + w_r r_A$ represents the total rate of incident data from $A$.

If we assume, say, that crime rate is uniform, so $\lambda = \lambda_A = \lambda_B$, then the probability of drawing a red ball has a limiting distribution equal to the Beta distribution that only depends on the initial number of red and black balls. Recall that the Beta distribution is the distribution on $[0, 1]$ given by probability distribution function

$$\frac{\Gamma(n_A)\Gamma(n_B)}{\Gamma(n_A + n_B)} x^{n_A - 1}(1 - x)^{n_B - 1},$$

where $\Gamma(x)$ is the usual gamma function, and $n_A$ and $n_B$ are the number of red and black balls respectively in the urn at the start. In particular, this means that the long-term probability of visiting an area is a random draw based on this initial data, and does not learn that the crime rates are equal. We refer to Ensign et al. (2018) for more complicated scenarios.

### A.3  PROACTIVE POLICING MODEL

In a proactive policing model, we have to make the following modification analogous to (Ensign et al., 2017, §5). Let
$$P_T = (A(t), Y(t), H(t), L(t))_{t=0}^T$$
be a partial monitoring problem up to time $T > 0$, where

- $A = A(t)$ is a set of actions,
- $Y = Y(t)$ a set of outcomes,
- $H : A \times Y \to \Sigma$ a feedback function that outputs some information or signal $\sigma \in \Sigma$ that the learner receives, and
- $L : A \times Y \to \mathbb{R}$ is a hidden loss function, usually taken to be positive.

For simplicity, we take $Y = \{y_A(t), y_B(t)\}$ to be the crime rate in areas $A$ and $B$ respectively at time $t$. Where before $A = \{a_A(t), a_B(t)\}$ is simply the allocation of a fixed number $k$ of police officers to $A$ and $B$ respectively, we now model $a_A(t)$ and $a_B(t)$ as probability distributions on $\{1, \ldots, k\}$ satisfying the condition
$$a_A(t) + a_B(t) = k$$
for all $t$. This amounts to the fact that if a proactive policing system is installed in $A$ rather than $B$ at time $t = t_0$ (under the rationale that the historical crime rate between $A$ and $B$ is nonuniform, say), then the probability that police are sent to $A$ is increased for all $t > t_0$.

A concrete example that a proactive policing system is not necessarily triggered by detected crime, we note the Project Green Light Detroit (PGLD) since 2016 that installs surveillance cameras at local businesses and other buildings, where not only are additional "virtual patrols" made, but also increased physical patrolling and priority 911 response is also provided (for a monthly fee and fixed startup cost) Circo & McGarrell (2021). This agrees with our model that increased policing ultimately occurs as a result of proactive policing, but it may be better to model it as a stochastic process to account for the variability of context. For specific applications, it will be useful to specify the $a_i(t)$ more explicitly.

### A.4  FAIRNESS PENALTIES

What are the algorithmic possibilities for fixing negative feedback loops in proactive policing? The analysis in Ensign et al. (2018) suggests that such loops are difficult to eliminate, whereas fairness penalties for Hawkes processes tend to require tradeoffs between fairness and accuracy Corbett-Davies et al. (2017); Shang et al. (2020) (see also Pastaltzidis et al. (2022); Alikhademi et al. (2022)). Ethically, the sacrifice of potentially accurate predictions might be understood as giving the benefit of the doubt (the separate problem of police violence aside).

More importantly, the policy implications of potential fairness implementations require external auditors of proactive policing to ensure accountability and neutrality, a difficult but necessary prospect in any implementation. On the other hand, a compartmental model of violence transmission Wiley et al. (2016) suggests that applying violence intervention methods to *all* police may be more effective. As such, the question of whether proactive policing can be debiased in practice is not a simple methodological issue but also a policy question in order to be made effective.

## B  IMPLEMENTATION USING SYNTHETIC DATA

We implemented our framework of proactive policing using synthetic data base on the urn model in Section A.2 as a simple special case of the partial monitoring problem $P_T$ described in Section A.3.

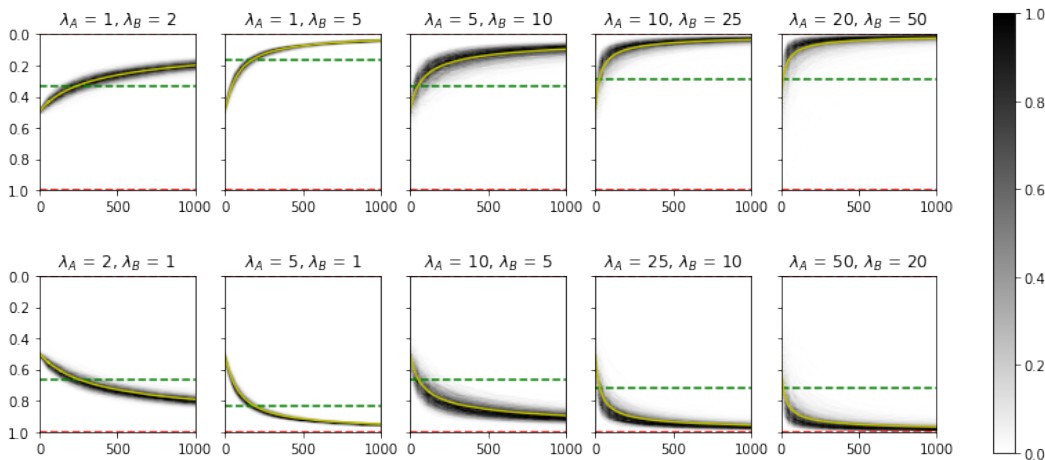

Figure 1: Distribution of policing resources deployed to region $A$ over 1,000 days versus the proportion of crime in region $A$ over a 1,000 days

In this setup, there are two locations $A$ and $B$ with nonuniform crime rates $\lambda_A$ and $\lambda_B$, iterated over 1000 days. We observe that in varying the rates in each location, runaway feedback loops continue to occur, moving away from the correct rate (green line), which is what the distribution of policing resources deployed (yellow line) should converge to if proactive policing were able to accurately approximate the actual crime rate. In particular, the latter would occur if proactive policing were to provide an efficient distribution of policing resources.

