# OpenReview forum: "Proactive policing as reinforcement learning"
_ICLR.cc/2023/TinyPapers — Submitted to Tiny Papers @ ICLR 2023_

### Official Review · Reviewer_gzXG · 2023-03-29

**Confidence:** 2

**Summary Of Contributions:**

The paper provides an expanded definition of proactive policing and relates it to predictive (or algorithmic) policing. The paper argues that models for predictive policing also extend to proactive policing in general, which opens several opportunities for new research and improvements for policing.

**Rating:**

Clear, Correct, and Reproducible (CCR): a submission which meets the reviewing criteria

**Strengths And Weaknesses:**

Strengths:

-	Provides a good overview of the predictive policing. The literature review provided a good account of historical models and criticisms for the practice.

-	The discussion section is strong and has several examples of exciting future research opportunities on this topic.

Weaknesses:

-	The contributions of the paper are not clearly indicated. While the paper focuses on the differences in definition between predictive policing and proactive policing, it’s not clear that the nuanced differences are upheld as notable within the community.

-	In particular, much of paper’s argument stems from a new definition of proactive policing. It is not clear from the paper if this would be a readily accepted definition in the community.


**Suggested Changes:**

-	The paper reads more like a research proposal than a research paper. Give more emphasis to new contributions.

-	More attention needs to be given to the new definition of proactive policing. Why have these methods not been considered for proactive policing previously? What is  the novelty of this new definition that would break the barriers for broader application of this research?

-	The claim that proactive policing leads to police violence is great topic for the discussion section, and the literature does seem to point to this as an interesting topic to study. However, there is not sufficient evidence provided in the paper to conclude that “proactive policing leas to over-policing, which in turn leads to the compounding effect of police violence.” Less confirmatory language should be used for discussion section topics.

---

### Official Review · Reviewer_VtNS · 2023-03-30

**Confidence:** 4

**Summary Of Contributions:**

The authors state that predictive policing algorithms: specifically the reinforcement learning algorithm by Ensign et al.(2018), could be adopted for proactive policing that goes beyond assessing the impact of predicting the allocation of police officers to police resources, including officers, gunshot detection mechanisms, society surveillance tools like facial recognition software, among others.

**Rating:**

Great Start (GS): a submission which meets some of the reviewing criteria but has room for improvement

**Strengths And Weaknesses:**

I very much like the angle the authors highlight in the paper. I strongly agree that predictive policing algorithms should collectively consider proactive policing. The generalization would help highlight other contributors to over and faulty predictive policing mechanisms and improve the assessment and evaluation of the schemes.

The authors go back and forth in a not very cohesive manner on the algorithmic shortcoming and improvement to getting rid of algorithmic assessment of policing. It is unclear what their main suggestion or key contribution is.
The paper looks to be in the very early stages of conception, and therefore, hard to assess and appreciate the contributions.

**Suggested Changes:**

The authors attempted to extend the predictive policing paper by Ensign et al.(2018). To improve the paper and highlight the contributions, the authors should complete the mathematical formulation on page 5 in the appendix.

Additionally, the authors should design experiments with their algorithm or framework on proactive policing data and show how their model compares to the predictive policing model.

Lastly, in the introduction/discussion sections, the authors could have a more focused or directed discussion to avoid confusion about their contributions or focus.

Suggestion: I think the authors should look into the causal inference perspective. I think it will provide interesting viewpoints on their work.

---

### Official Review · Reviewer_pJnE · 2023-03-31

**Confidence:** 4

**Summary Of Contributions:**

In this paper, the authors focus on the problem of proactive policing. To this end, this paper proposes that the mathematical models of predictive policing can be adapted to proactive policing. Therefore, proactive policing can be framed as a reinforcement learning problem.

**Rating:**

Great Start (GS): a submission which meets some of the reviewing criteria but has room for improvement

**Strengths And Weaknesses:**

**Strength**
* S1: The paper clearly explains the predictive policing as well as the mathematical models associated with it.
* S2: Sufficient mathematical proof has been provided

**Weaknesses**
* W1: There seems to be a gap in explanation of the topic. Police violence has been discussed in Section 3.1 but the motivation for relating it to the topic has not been explained.
* W2: No models have been trained to verify the claims in the paper



**Suggested Changes:**

* Some sentences should be added to explain the motivation to discuss police violence (as a form of negative feedback)
* Some experiments should be conducted with existing data to verify that the mathematical models of predictive policing do apply to proactive policing

---

### Author Response · Authors · 2023-05-30
**Response to reviewers/Archival opt in**

We wish to opt-in for archival.

We are grateful to the reviewers for helpful feedback and suggestions improving the paper. Responses to reviewer sugestions:

A. Reviewer pJnE

1. Additional clarification has been added in the introduction to explain the motivation to discuss police violence as a form of negative feedback, including public health implications.

2. Experiments have been conducted with synthetic data to verify that the mathematical models of predictive policing do apply to proactive policing. We use the Polya urn model as employed in the Ensign et al. paper which we discuss, added in Appendix B.

B. Reviewer VtNS

1. A more complete mathematical formulation on page 5 in the appendix has been provided.

2. As discussed in (A2) above, we design a simple experiment with our framework on proactive policing data and show how the model is indeed consistent with the feedback loops generated in the predictive policing model.

3. In the introduction/discussion sections, we have aimed to provide a more focused discussion to avoid confusion about our contributions.

4. We are grateful for the suggestion on the causal inference perspective. It is indeed an interesting viewpoint which we hope to consider in the near future.

Reviewer gzXG

1. As in (A1) and (B3), the introduction and discussion have been clarified to give more emphasis to new contributions.

2. We have tried to give aditional indications as to why these methods have not been considered for proactive policing previously and what is the novelty of this new definition that would break the barriers for broader application of this research, but for reasons of space we are not able to elaborate this in much detail.

3. We have rephrased the paper to use less confirmatory language regarding the connection between proactive policing, over-policing, and police violence.

---

### Meta-Review · Area_Chair_T8fC · 2023-04-02

**Recommendation:** Invite to revise
**Confidence:** 4

**Metareview:**

This is a timely paper that aims to tackle algorithmic and policing fairness, and bring new ideas and techniques in to the discourse.  The paper is reasonably well motivated, and is original enough in intent.  Unfortunately, the reviewers had issues with the clarity of the text, and that the proposed method itself was not clearly explained, or tested or explored experimentally.  I firmly believe future iterations of this work could be significant and a good contribution to either a TinyPapers style workshop, or at a more specialised fairness workshop/conference.

**Summary:**

This paper discusses links between algorithmic approaches to policing and implicit biases, and how these can be re-considered under an RL framework. All three reviewers agreed that the topic and paper are a fantastic start to an area with great potential.  However, no empirical results are presented to justify or help explain the proposed methods, and the text itself could benefit from some more rounds of revision to make the method clearer and highlight the contributions.

**Comments And Feedback To The Authors:**

All three reviewers gave good, constructive feedback on how to improve the paper.

In lieu of applying a method to real data, it would be great to see a synthetic example, or example on a limited dataset, to really highlight what the proposed method brings.

I encourage the authors to keep working on this project, and maybe seek out a policing/fairness-specific workshop to submit to, to gather more domain-specific feedback, collaborations, ideas and data.

**Reason For Not Giving A Higher Recommendation:**

All three reviewers agreed that the work is not quite ready for publication yet, but that it shows great promise.  Revisions to the text are required, and some empirical validation would greatly strengthen the work.

**Reason For Not Giving A Lower Recommendation:**

N/A

---

### Decision · Program_Chairs · 2023-04-08

Revision accepted; invite to archive